# Pregnant women with mild COVID-19 followed in community setting by telemedicine, and factors associated with unfavorable outcome

**Aurélien Dinh**[1]*, **Florian Drouet**[2], **Agnes Dechartres**[3,4], **Youri Yordanov**[5], **Clara Duran**[1], **Nicolas Schmidt**[2], **Amélie Banzet**[2], **Marie-Hermine Perrier**[2], **Nathalie Mosquet**[2], **François-Xavier Lescure**[6], **Patrick Jourdain**[7], **Jacky Nizard**[8], **Xavier Masingue**[2], on behalf of the AP-HP/Universities/Inserm COVID-19 research collaboration**¶**

1 Infectious Disease, Raymond-Poincaré University Hospital APHP, Garches, France, 2 Covidom Regional Telemedicine Platform, APHP, Great Paris area, France, 3 Institut Pierre Louis d'Epidémiologie et de Santé Publique, INSERM, Sorbonne Université, AP-HP, Paris, France, 4 Département de Santé Publique, Hôpital Pitié Salpêtrière, Sorbonne Université, Paris, France, 5 Emergency Department, University Hospital Saint Antoine, APHP, Paris, France, 6 Infectious Disease, Bichat University Hospital APHP, Paris, France, 7 Cardiology Department, University Hospital of Kremlin Bicêtre, APHP, Le Kremlin Bicêtre, France, 8 Department of Obstetrics and Gynecology, Pitié-Salpétrière University Hospital, Sorbonne Université, APHP, Paris, France

¶ Membership of the AP-HP/Universities/Inserm COVID-19 research collaboration is listed in the Acknowledgments.
* aurelien.dinh@aphp.fr

⛔ OPEN ACCESS

**Data Availability Statement:** Data privacy regulations prohibit deposition of individual level data to public repositories and the ethical approval

## Abstract

### Objectives

Few is known on pregnant women with mild COVID-19 managed in a community setting with a telemedicine solution, including their outcomes. The objective of this study is to evaluate the adverse fetal outcomes and hospitalization rates of pregnant COVID-19 outpatients who were monitored with the Covidom© telemedicine solution.

### Methods

A nested study was conducted on pregnant outpatients with confirmed COVID-19, who were managed with Covidom© between March and November 2020. The patients were required to complete a standard medical questionnaire on co-morbidities and symptoms at inclusion, and were then monitored daily for 30 days after symptom onset. Adverse fetal outcome was defined as a composite of preterm birth, low birthweight, or stillbirth, and was collected retrospectively through phone contact with a standardized questionnaire.

### Results

The study included 714 pregnant women, with a median age of 32.0 [29.0–35.0] and a median BMI of 23.8 [21.3–27.0]. The main comorbidities observed were smoking (53%), hypertension (19%). The most common symptoms were asthenia (45.6%), cough (40.3%) and headache (25.7%), as well as anosmia (28.4%) and agueusia (32.3%). Adverse fetal

does not cover public sharing of data for unknown purposes. Upon contact with the authors or at guichet.industriel.drc@aphp.fr an institutional data transfer agreement may be established, and data shared if the aims of data use are covered by ethical approval and patient consent.

**Funding:** This study received a funding by the Programme Hospitalier de Recherche Clinique 2020 of the French Ministry of Health, by a research fund by APHP-Fondation de France. The Covidom platform received a funding from EIT Health specific Covid-19 fund. The funders play no role in the study design, data collection and analysis, decision to publish, or preparation of the manuscript.

**Competing interests:** The authors have declared that no competing interests exist.

outcomes occurred in 64 (9%) cases, including 38 (5%) preterm births, 33 (5%) low birth-weights, and 6 (1%) stillbirths. Hospitalization occurred in 102 (14%) cases and was associated with adverse fetal outcomes (OR 2.4, 95% CI 1.3–4.4).

## Conclusions

Our study suggests that adverse fetal outcomes are rare in pregnant women with mild COVID-19 who are monitored at home with telemedicine. However, hospitalization for COVID-19 and pregnancy-induced hypertension are associated with a higher risk of adverse fetal outcome.

## Introduction

COVID-19 rapidly emerged as a global pandemic just a few months after its onset in December 2019 [1]. It has created an urgent need for innovative ways to deliver healthcare services to patients while minimizing the risk of disease transmission. Telemedicine has emerged as a safe and effective alternative to in-person consultations, enabling patients and healthcare professionals to connect remotely and exchange information without risking viral exposure [2].

Teleconsultations have proven particularly useful for suspected COVID-19 cases, allowing healthcare professionals to assess and monitor patients remotely while guiding them through the diagnostic and treatment process [3]. By using telemedicine technologies, healthcare professionals could provide patients with virtual access to a variety of healthcare services, including medical consultations, remote patient monitoring, and prescription management. These services have been particularly important for individuals with pre-existing medical conditions who are at higher risk of developing complications from COVID-19.

Indeed, several risk factors associated with severe outcomes have been well identified, including age, obesity, immunodepression, and chronic lung disease [4]. Pregnant women with severe COVID-19 have a high risk of fetal adverse events [5, 6], and pregnancy is a risk factor for severe disease [7, 8]. However, little is known about pregnant women with mild COVID-19 in the community setting.

Few is known on telemedicine during COVID-19 among pregnant women and outcome of this specific population with mild COVID-19.

Our study aims to describe a cohort of pregnant women with mild COVID-19 managed in a community setting with a telemedicine solution, including their outcomes in terms of adverse fetal events and hospital admission for COVID-19, as well as factors associated with such outcomes.

## Methods

This study used the Covidom© cohort, a prospective cohort of ambulatory COVID-19 patients in the greater Paris area, who used the Covidom© telemedicine solution for home monitoring [3]. The Covidom© solution consists of a user-friendly web application for patients and a regional control center system that manages alerts. Patients register with a physician's help and complete daily self-administered monitoring questionnaires for 30 days after symptom onset. Patient answers are classified into four categories by an automated algorithm: No alert, Orange alert, Red alert, and Grey alert.

The Covidom regional control center operates from 8 a.m. to 8 p.m., 7 days a week, and handles alerts generated by patient answers. Trained responders and a supervising physician

handle the alerts, offering solutions such as medical advice, referral to their GP, or hospitalization [3].

All pregnant women with a positive COVID-19 PCR during pregnancy, registered in Covidom as an outpatient between March 10th, 2020 and November 30th, 2020, were included in this study if they completed the Covidom medical questionnaire and agreed to participate. Patient characteristics, including medical conditions prior to and during pregnancy, active smoking, and symptoms at inclusion, were collected through a self-reported medical questionnaire. Data on delivery, postpartum, and neonatal outcomes were collected retrospectively through phone contact with a standardized questionnaire. Adverse fetal outcome was defined as a composite of preterm birth (<37 weeks of gestation), low birthweight (< 2500g) or stillbirth/perinatal death. We also evaluated hospital admissions for worsening of COVID-19.

### Statistical analysis

Continuous variables were presented as median and interquartile range (IQR), and categorical variables as numbers (percentages). Baseline characteristics were compared using t-tests and chi-square tests for continuous and categorical variables, respectively. Univariate analysis using odds ratios (OR) and 95% confidence intervals (CI) was conducted to evaluate factors associated with adverse fetal outcomes and COVID-19 hospital admissions. Python was used for the analysis.

### Ethical committee

Patients provided electronic consent for the Covidom telesurveillance program and were informed of the use of their anonymized data for research. Written informed consent was waived by the ethics committee. The study was approved by the Scientific and Ethical Committee of AP-HP (IRB00011591).

### Results

In the Covidom cohort, 1613 pregnant women were included as outpatients, out of which 714 (44.3%) agreed to participate and provided data on their delivery and neonate. The median age of participants was 32.0 [IQR 29.0–35.0] years old, with a median BMI of 23.8 [IQR 21.3–27.0]. The main comorbidity reported was smoking (7.4%), while gestational diabetes mellitus was present in 15.4% of cases. The most common symptoms were fatigue (45.7%), cough (40.3%), and headache (25.7%), as well as anosmia (28.4%), and agueusia (32.3%). Table 1 presents the demographic and baseline characteristics of the participants.

Adverse fetal outcomes occurred in 64 (9%) cases, which included 38 (5%) preterm births, 33 (5%) low birthweights, and 6 (1%) stillbirths. Three premature deaths were observed at 15 (due to cardiac arrest), 22 (due to antiphospholipid syndrome with multiple infarctions), and 24 (unexplained) weeks of gestation, as well as one case of aspiration pneumonia from amniotic fluid at birth, one case of placental abruption, and one case of severe intrauterine growth restriction.

The only risk factor associated with adverse fetal outcome was pregnancy-induced hypertension (OR = 2.8; 95% CI: 1.3–5.9). The median period of catching COVID-19 was 24 weeks of pregnancy, and it was similar between groups with favorable and adverse fetal outcomes.

Hospital admission due to COVID-19 occurred in 102/714 (14%) women. Factors associated with hospital admission due to COVID-19 included chronic hypertension (OR = 3.6, 95% CI: 1.4–9.5) and symptoms such as conjunctivitis, anorexia, chest tightness, dyspnea, myalgia, fever, nausea/vomiting, cough, diarrhea, chest pain, and chilblains (as shown in Table 2). Women who were hospitalized for COVID-19 were more likely to have adverse fetal outcomes than those who were not admitted (OR = 2.4, 95% CI 1.3–4.4).

**Table 1. Adverse fetal outcome in women with COVID-19 in community setting.**

| | Adverse fetal outcomes (N = 64) | No adverse fetal outcome (N = 650) | P-value |
|---|---|---|---|
| **Characteristics** | | | |
| Age (year, median [IQR]) | 34 [29–36] | 32 [29–35] | 0.376 |
| BMI (median [IQR]) | 24 [22–28] | 24 [21–27] | 0.471 |
| Weeks of amenorrhea at inclusion (median [IQR]) | 24 [18–28] | 24 [15–31] | 0.850 |
| Age > 35 | 17 (27) | 158 (24) | 0.689 |
| BMI > 30 | 12 (19) | 91 (14) | 0.302 |
| **Comorbidities** | | | |
| Cardiopathy | 1 (2) | 13 (2) | 0.810 |
| Renal failure | 0 (0) | 3 (0) | 0.586 |
| Hypertension | 3 (5) | 16 (2) | 0.291 |
| Diabetes mellitus | 2 (3) | 7 (1) | 0.161 |
| Smoking | 5 (8) | 48 (7) | 0.901 |
| **Comorbidities associated with pregnancy** | | | |
| Pregnancy cholestasis | 2 (3) | 11 (2) | 0.413 |
| Gestational diabetes mellitus | 14 (22) | 96 (15) | 0.161 |
| Pregnancy-induced hypertension | 10 (16) | 40 (6) | <0.01 |
| **Symptoms** | | | |
| COVID-19 duration (day, median [IQR]) | 12 [7–18] | 14 [7–15] | |
| Headaches | 16 (25) | 168 (26) | 0.883 |
| Dyspnea | 15 (23) | 144 (22) | 0.814 |
| Fatigue | 24 (38) | 302 (46) | 0.170 |
| Cough | 28 (44) | 260 (40) | 0.560 |
| Fever | 16 (25) | 186 (29) | 0.540 |
| Chills | 10 (16) | 146 (22) | 0.207 |
| Agueusia | 20 (31) | 210 (32) | 0.863 |
| Anosmia | 19 (30) | 184 (28) | 0.815 |
| Myalgia | 10 (16) | 69 (11) | 0.223 |
| Chest pain | 7 (11) | 65 (10) | 0.812 |
| Diarrhea | 6 (9) | 79 (12) | 0.512 |
| Nausea/vomiting | 13 (20) | 121 (19) | 0.740 |
| Chest tightness | 6 (9) | 36 (6) | 0.213 |
| Anorexia | 8 (12) | 83 (13) | 0.951 |
| Conjunctivitis | 2 (3) | 8 (1) | 0.219 |
| Chilblains | 1 (2) | 2 (0) | 0.139 |
| Skin rash | 0 (0) | 12 (2) | 0.273 |
| **Outcome** | | | |
| Child weight (g, median [IQR]) | 2455 [2200–3300] | 3300 [3200–3500] | <0.01 |
| Covid hospitalization | 17 (30) | 79 (14) | <0.01 |
| Child death | 6 (11) | 0 (0) | <0.01 |

Data are n (%) unless otherwise stated. BMI: body mass index; IQR: interquartile range

## Discussion

In our cohort of pregnant women with mild COVID-19, followed in a community setting by telemedicine (Covidom© cohort), we observed a 9% rate of adverse fetal outcomes. Pregnancy-induced hypertension and hospitalization for COVID-19 were the only factors significantly associated with adverse fetal outcomes.

**Table 2. Hospitalization among pregnant women with COVID-19 in community setting.**

| | COVID-19 non-Hospitalization (N = 612) | COVID-19 Hospitalization (N = 102) | P-value |
|---|---|---|---|
| **Characteristics** | | | |
| Age (year, median [IQR]) | 32 [29–35] | 32 [29–35] | 0.794 |
| BMI (median [IQR]) | 24 [21–27] | 25 [22–29] | <0.01 |
| Weeks of amenorrhea at inclusion (median [IQR]) | 24 [14–30] | 28 [24–35] | <0.01 |
| Age > 35 | 150 (25) | 25 (25) | 1.0 |
| BMI > 30 | 84 (14) | 19 (19) | 0.192 |
| **Comorbidities** | | | |
| Cardiopathy | 12 (2) | 2 (2) | 1.0 |
| Renal failure | 3 (0) | 0 (0) | 0.479 |
| Hypertension | 12 (2) | 7 (7) | <0.01 |
| Diabetes mellitus | 8 (1) | 1 (1) | 0.784 |
| Smoker | 50 (8) | 3 (3) | 0.062 |
| **Comorbidties associated with pregnancy** | | | |
| Pregnancy cholestasis | 11 (2) | 2 (2) | 0.909 |
| Gestational diabetes mellitus | 92 (15) | 18 (18) | 0.498 |
| Pregnancy-induced hypertension | 43 (7) | 7 (7) | 0.952 |
| **Symptoms** | | | |
| COVID-19 duration (day, median [IQR]) | 14 [7–15] | 15 [10–21] | 0.151 |
| Headaches | 160 (26) | 24 (24) | 0.576 |
| Dyspnea | 120 (20) | 39 (38) | <0.01 |
| Fatigue | 271 (44) | 55 (54) | 0.070 |
| Cough | 233 (38) | 55 (54) | <0.01 |
| Fever | 157 (26) | 45 (44) | <0.01 |
| Chills | 124 (20) | 32 (31) | 0.012 |
| Agueusia | 197 (32) | 33 (32) | 0.974 |
| Anosmia | 168 (27) | 35 (34) | 0.155 |
| Myalgia | 58 (9) | 21 (21) | <0.01 |
| Chest pain | 56 (9) | 16 (16) | 0.042 |
| Diarrhea | 66 (11) | 19 (19) | 0.024 |
| Nausea and vomiting | 103 (17) | 31 (30) | <0.01 |
| Chest tightness | 30 (5) | 12 (12) | <0.01 |
| Anorexia | 64 (10) | 27 (26) | <0.01 |
| Conjunctivitis | 6 (1) | 4 (4) | 0.019 |
| Chilblains | 0 (0) | 3 (3) | <0.01 |
| Skin rash | 10 (2) | 2 (2) | 0.812 |
| **Outcome** | | | |
| Child weight (g, median [IQR]) | 3330 [3200–3500] | 3300 [2900–3438] | <0.01 |
| Childbirth's week (median [IQR]) | 40 [39–40] | 39 [38–40] | <0.01 |
| Child death | 5 (1) | 1 (1) | <0.01 |

Data are n (%) unless otherwise stated. BMI: body mass index; IQR: interquartile range

The COVID-19 pandemic has underscored the importance of telemedicine in enabling communication between patients and healthcare professionals, especially when in-person consultations were not feasible or posed a risk of viral transmission or quarantine requirements [2]. Teleconsultations provided a secure and efficient method for evaluating suspected COVID-19 cases, facilitating diagnosis and treatment while reducing the risk of disease

transmission [3]. Telemedicine also enabled the continuation of several essential clinical services during COVID-19, especially those at high risk of unfavorable outcomes or specific populations such as pregnant women, as shown with the Covidom© solution [3].

Surprisingly, our population of women with mild COVID-19 seems to have a similar rate of adverse fetal outcomes to that of the general population, as per national epidemiological data [9, 10]. In contrast, most meta-analyses and systematic reviews of studies on pregnancy and COVID-19 focused on severe COVID-19 forms and found an increased risk of adverse birth outcomes among pregnant women with SARS-CoV-2 infection compared to those without SARS-CoV-2 infection [5, 6, 11, 12].

However, our study found that mild COVID-19 managed in a community setting had a low impact on adverse fetal outcomes. Our results are consistent with several large cohort studies that showed SARS-CoV-2 infection during pregnancy was not associated with adverse birth outcomes or unfavorable maternal outcomes [13, 14].

Despite the increasing number of published studies on COVID-19 in pregnancy, there are still insufficient good-quality data to draw firm and unbiased conclusions regarding specific complications of COVID-19 in pregnant women. Our results suggest that the severity of the COVID-19 episode is a major risk factor for adverse fetal outcomes, as hospitalization due to COVID-19 was found to be an associated factor.

In our study, the other risk factor associated with adverse fetal outcomes was pregnancy-induced hypertension, which should lead to increased monitoring among pregnant women with COVID-19 if hypertension occurs.

At last, in our cohort, the risk factors and symptoms for COVID-19 hospitalization were similar to those in the general population, including hypertension, chest tightness, dyspnea, myalgia, and fever. Previous research has indicated that the risk factors for severe COVID-19 are similar among pregnant and nonpregnant women [7, 8].

However, despite the low rate of adverse fetal outcomes in our cohort study, vaccination among pregnant women is still highly recommended to prevent severe COVID-19. A study from the US showed that vaccination of pregnant women decreased the risk of hospitalization of the newborn by 60%, and 88% of newborns in the ICU had unvaccinated mothers [15].

Our study had several limitations, including self-reporting by patients resulting in possible recall bias, non-respondents potentially differing from respondents regarding pregnancy outcomes, and a lack of comparative data. Therefore, further studies are needed to confirm our findings.

## Conclusion

In our study of pregnant women with mild COVID-19 in a community setting who received telemedicine follow-up, the incidence of adverse fetal outcomes was low. Our findings indicate that pregnancy-induced hypertension and hospitalization for COVID-19 were associated with an increased risk of adverse fetal outcomes. Therefore, vaccination and close monitoring are essential, particularly for pregnant women with pregnancy-induced hypertension.

## Acknowledgments

The authors would like to thank Amira BOUROKBA, Estelle HORISZNY, Mouna LABIB, and Céline MECHEMACHE for their help during the follow-up of the pregnant patients included in the COVIDOM solution.

AP-HP / Universities / Inserm COVID-19 research collaboration members. Writing committee: Dinh Aurélien (Infectious Disease Department, University Hospital Raymond-Poincaré, Assistance Publique–Hôpitaux de Paris, Garches, Paris Saclay University, Garches,

France; lead author– aurelien.dinh@aphp.fr); Mercier Jean-Christophe, Artigou Jean-Yves, Juillière Yves (COVIDOM, Assistance Publique–Hôpitaux de Paris, Paris, France); Jaulmes Luc (Centre de PharmacoÉpidémiologie (Cephepi), Pitié Salpêtrière Hospital, Paris, France); Yordanov Youri (Emergency Department, University Hospital Saint-Antoine, Assistance Publique–Hôpitaux de Paris, Sorbonne University, Paris, France); Jourdain Patrick (Cardiology Department, University Hospital Bicêtre, Assistance Publique–Hôpitaux de Paris, Paris Saclay University, Le Kremlin-Bicêtre, France); Data-sciences committee: Apra Caroline (Sorbonne Université, AP-HP, Hôpital Pitié Salpêtrière, Service deNeurochirurgie, Paris, France); Jaulmes Luc (Centre de Pharmaco-Épidémiologie (Cephepi), Pitié Salpêtrière Hospital, Paris, France); Mensch Arthur (Ecole Normale Supérieure, PSL University, CNRS, Départment de Mathématiques et Applications, Paris, France); Scientific committee: Aime-Eusebi Amélie, Apra Caroline, Bleibtreu Alexandre, Debuc Erwan, Dechartres Agnes, Deconinck Laurene, Dinh Aurelien, Jourdain Patrick, Katlama Christine, Lebel Josselin, Lescure François-Xavier, Yordanov Youri; Covidom regional center steering commitee: Artigou Yves, Banzet Amelie, Boucheron Elodie, Boudier Christiane, Buzenac Edouard, Chapron Marie-Claire, Chekaoui Dalhia, De Bastard Laurent, Debuc Erwan, Dinh Aurelien, Grenier Alexandre, Haas Pierre-Etienne, Hody Julien, Jarraya Michele, Jourdain Patrick, Lacaille Louis, Le Guern Aurelie, Leclert Jeremy, Male Fanny, Marchand-Arvier Jerome, Martin-Blondet Emmanuel, Nassour Apolinne, Ourahou Oussama, Penn Thomas, Ribardiere Ambre, Robin Nicolas, Rouge Camille, Schmidt Nicolas, Villie Pascaline.

## Author Contributions

**Conceptualization:** Aurélien Dinh, Agnes Dechartres, Xavier Masingue.

**Data curation:** Aurélien Dinh, Florian Drouet, Agnes Dechartres, Youri Yordanov, Clara Duran, Nicolas Schmidt, Nathalie Mosquet.

**Formal analysis:** Aurélien Dinh, Florian Drouet, Agnes Dechartres, Youri Yordanov, Clara Duran, Nicolas Schmidt, Nathalie Mosquet.

**Investigation:** Aurélien Dinh, Florian Drouet, Agnes Dechartres, Youri Yordanov, Clara Duran, Nicolas Schmidt, Nathalie Mosquet.

**Writing – original draft:** Aurélien Dinh, Florian Drouet, Agnes Dechartres, Clara Duran.

**Writing – review & editing:** Aurélien Dinh, Florian Drouet, Agnes Dechartres, Youri Yordanov, Clara Duran, Nicolas Schmidt, Amélie Banzet, Marie-Hermine Perrier, Nathalie Mosquet, François-Xavier Lescure, Patrick Jourdain, Jacky Nizard, Xavier Masingue.

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
