## [Decision Letter · Decision Letter 0]

2 May 2023

PONE-D-22-18040Pregnant women with mild COVID-19 followed in community setting by telemedicine, and factors associated with unfavorable outcomePLOS ONE

Dear Dr. Dinh,

Thank you for submitting your manuscript to PLOS ONE. After careful consideration, we feel that it has merit but does not fully meet PLOS ONE’s publication criteria as it currently stands. Therefore, we invite you to submit a revised version of the manuscript that addresses the points raised during the review process.

We look forward to receiving your revised manuscript.

Kind regards,

Burak Bayraktar

Academic Editor

PLOS ONE

4. One of the noted authors is a group or consortium [the AP-HP/Universities/Inserm COVID-19 research collaboration]. In addition to naming the author group, please list the individual authors and affiliations within this group in the acknowledgments section of your manuscript. Please also indicate clearly a lead author for this group along with a contact email address

Additional Editor Comments:

1. Introduction is short. Expand this section a bit.

In addition;

Provide information about telemedicine in one paragraph, as well as mention its uses and why it is appropriate in the COVID era. You can talk about the advantages of telemedicine in this period.

2. “Covidom includes a web application sending daily questionnaires to patients and a regional control center dealing with alerts generated by the response to questionnaires” How to make covidom, talk a little more.

3. Separate the statistical analysis paragraph as a subtitle: Statistical Analysis

4. “Child death 6 (11)” Include indications of death where possible.

5. Add Table 2 to results section (no discussion)

6. “Thus, surprisingly, our population of women with mild COVID-19 seems to have around the same rate of fetal adverse outcome than the general population” How is this situation in the literature?

7. “In our study, the other risk factor associated with adverse fetal outcome was pregnancy- induced hypertension. These results should lead to increase monitoring among pregnant women with COVID-19 if hypertension occurs.” Any ideas on the mechanism? Similar results are available in the literature. What do you think?

8. Do not analyze each research result in a separate paragraph in the Discussion section. Make a single paragraph of conclusions on similar topics.

9. Can the results regarding the periods of catching COVID 19 (first-second and third trimester) of pregnant women change? I guess you didn't group according to trimesters.

10. Indicate that you are only including mild cases of COVID 19 in the material method, where appropriate.

11. I did not see information about the vaccination status in your population.

12. Discussion section should be more organized.

Reviewers' comments:

Reviewer's Responses to Questions

**Comments to the Author**

1. Is the manuscript technically sound, and do the data support the conclusions?

Reviewer #1: Yes

Reviewer #2: Partly

Reviewer #3: Yes

Reviewer #4: Yes

Reviewer #5: Yes

2. Has the statistical analysis been performed appropriately and rigorously? 

Reviewer #1: Yes

Reviewer #2: Yes

Reviewer #3: Yes

Reviewer #4: Yes

Reviewer #5: Yes

3. Have the authors made all data underlying the findings in their manuscript fully available?

Reviewer #1: Yes

Reviewer #2: Yes

Reviewer #3: Yes

Reviewer #4: Yes

Reviewer #5: Yes

4. Is the manuscript presented in an intelligible fashion and written in standard English?

Reviewer #1: Yes

Reviewer #2: Yes

Reviewer #3: Yes

Reviewer #4: Yes

Reviewer #5: Yes

5. Review Comments to the Author

Reviewer #1: A very well done project.

I would suggest adding information related to other perinatal outcomes that have already been shown to increase in prevalence as a sequela of COVID infection, e.g., preeclamsia (early/tardia), placental abruption.

Reviewer #2: 1. Was there cross over from those who consented to telemedicine to in-person consult? From the results, it seemed that 100% of the participants had remained on telemedicine follow up. If so, probably can describe in more detail in the discussion why you were able to keep them well managed on telemedicine. How did you pick up the worsening of symptoms. Were you able to remotely measure parameters or examine the patient?

2. Besides the COVID-19 follow up, was the Ob/Gyn follow up also on telemedicine/hybrid or it was purely in person?

3. From what i understand, there is a pretty large group 55.7% (pregnant with mild COVID-19) that was on follow up with the health system but did not consent to telemedicine follow up. I thought it would have been useful to compare the 44.3% that had underwent telemedicine guided by the daily self-reporting of symptoms and whether it made a difference compared to those who only came for in-person follow ups when they were unwell, or as a routine.

4. Specifically for PIH, it did show a significant difference for adverse outcomes as defined by the study. It was not clear COVID-19 worsened the incidence of adverse outcomes or not. From my limited understanding, even without COVID-19 infection, those with PIH could have pre-eclampsia leading to SGA, IUGR, indicated pre-term deliveries already.

Reviewer #3: Thank you for your wonderful efforts and hard work that is clearly reflected in your project.

I may highlight these points, not to decline your work, but to have a better view of the project outcomes.

I am happy to reconsider your work after making some clarifications on these points:

1- Correlation between the time that the patient had COVID-19 and her gestational age? and was there any difference between different gestational ages and the outcome?

2- When you describe mild COVID-19, can you please identify all the criteria and clinical symptoms that made you consider it as mild COVID?

3- Duration between having COVID and the outcome. i.e: Can you highlight timelines? if the patient had COVID-19 (mild symptoms) when exactly she either delivered, had a preterm delivery, or had a stillbirth?

4- What do you mean by chest oppression in lines 97 & 138?

5 - Have the patients who had PIH, developed it after COVID-19 or before?

6 - Any of these ladies got vaccinations or not?

7- Any other factors that was contributing to the outcome? for example: when you mentioned stillbirth or preterm, do you know if any previous obstetric history of the same issues?

8- Any ultrasound or Doppler studies done for these patients, either before, during, or after having COVID, that identified any fetal issues?

Reviewer #4: This manuscript seems smart and clear with several meaningful tables. To my disappointed, these data are obtained from self-reported informations, which have many limitations.

For example, in case that pregnant women suffering from COVID-19 infection requires some medications, the kind of drug and the dose may be more influenced to the fetal outcomes.

After reading this manuscript, I have two requests shown as follows;

#1. Although statistic significant difference are shown undoubtedly, the number of fetal death seems much close between hospitalization group (1/102) and non-hospitalization one (5/612). The rate of pregnancy-induced hypertension are not significant differences between these two groups, so I would like you to disclose more information about these six cases such as weeks of delivery and the existence of fetal distress.

#2. In discussion you mentioned the relation between the ICU admission of newborn children and the vaccination of COVID-19. We must desire to know about the relation in this Covidom research. If you have already evaluated the relation, please let us show that.

Reviewer #5: 1. ABSTRACT: The abstract should start with a brief background to the study highlighting what is known or unknown about the proposed study.

Objective: The word “we” should be removed from the objective

Methods: The information doesn’t convey succinctly how the study was done.

Conclusion: Conclusion should also be made on the role or impact of telemedicine

2. The introduction is too short and doesn’t state the extent of the problem neither does it contain the justification for the study. A more elaborate introduction should be written.

3, The methodology should describe in details what the COVIDOM telemedicine software is all about. How does it work? What version was used for this study? Is it a video conferencing or audio-conferencing app? This information should be clearly spelt out in the methodology

4. The methodology should describe in details how the telemedicine was used for the study. How were the questionnaires delivered? Was it delivered to their phones or email? Was there an online/web based face-to-face interview?

5 Lines 114,115,116-- Pregnant women with severe COVID-19 had higher risk of preeclampsia, preterm birth, gestational diabetes, and low birthweight than pregnant women with mild disease. This outcome is expected.

6. Table 2 should be moved to the result section

7. The study involved the use of telemedicine. Was there any advantage regarding the use of telemedicine over other studies that telemedicine was not used? There should be a robust discussion on telemedicine.

8. The references were not properly written. Almost all the references still have the DOI numbers. They should be rewritten according to Vancouver guidelines.

6. PLOS authors have the option to publish the peer review history of their article (what does this mean?). If published, this will include your full peer review and any attached files.

Reviewer #1: **Yes: **Borboa Olivares Hector

Reviewer #2: No

Reviewer #3: **Yes: **Mena Abdalla

Reviewer #4: No

Reviewer #5: No

---

## [Author Response · Author response to Decision Letter 0]

23 May 2023

Paris, May 23, 2023

Dear Editor,

Thank you for offering us the opportunity to submit a revised version of our manuscript “Pregnant women with mild COVID-19 followed in community setting by telemedicine, and factors associated with unfavorable outcome”. 

We have carefully considered yours and the mostly helpful comments from the reviewers, and provide below a point-by-point response to these comments, as well as to your comments, with changes made in the manuscript indicated where appropriate.

Kind regards

Aurélien Dinh, MD

Editor's comments:

1. Introduction is short. Expand this section a bit.

In addition;

Provide information about telemedicine in one paragraph, as well as mention its uses and why it is appropriate in the COVID era. You can talk about the advantages of telemedicine in this period.

We thank the reviewer for allowing us the opportunity to include more information about telemedicine, especially during COVID-19, and its potential advantages. We have added one paragraph in the introduction and one in the discussion section of the manuscript.

2. “Covidom includes a web application sending daily questionnaires to patients and a regional control center dealing with alerts generated by the response to questionnaires” How to make covidom, talk a little more.

We thank the reviewer for their feedback and have included a detailed description of the COVIDOM solution and the remote monitoring process in the Materials and Methods section.

3. Separate the statistical analysis paragraph as a subtitle: Statistical Analysis

We have separated this paragraph.

4. “Child death 6 (11)” Include indications of death where possible.

We have added the indications of death in the results section.

5. Add Table 2 to results section (no discussion)

We thank the reviewer to have highlight this mistake. We have added the table 2 to the result part.

6. “Thus, surprisingly, our population of women with mild COVID-19 seems to have around the same rate of fetal adverse outcome than the general population” How is this situation in the literature?

We have reworded this part and added recent data from literature.

7. “In our study, the other risk factor associated with adverse fetal outcome was pregnancy- induced hypertension. These results should lead to increase monitoring among pregnant women with COVID-19 if hypertension occurs.” Any ideas on the mechanism? Similar results are available in the literature. What do you think?

Pregnancy-induced hypertension is a well-known risk factor for both premature death and complicated pregnancy. It has been extensively reported in the literature.

8. Do not analyze each research result in a separate paragraph in the Discussion section. Make a single paragraph of conclusions on similar topics.

We reworded this paragraph accordingly.

9. Can the results regarding the periods of catching COVID 19 (first-second and third trimester) of pregnant women change? I guess you didn't group according to trimesters.

The median time of COVID-19 onset was 24 weeks of pregnancy, and this duration was similar between the group with favorable and unfavorable outcomes. We have incorporated this information into the manuscript text.

10. Indicate that you are only including mild cases of COVID 19 in the material method, where appropriate.

We thank the reviewer for allowing us to clarify this information. Indeed, the patients included in our study were only from the community setting, and therefore had mild forms of COVID-19.

11. I did not see information about the vaccination status in your population.

Most of the patients were unvaccinated as the study was conducted during the first three surges of the pandemic when the vaccine was not yet available in France. Therefore, it can be considered that the study population was not vaccinated.

12. Discussion section should be more organized.

We strongly agree with the reviewer's comments and have created a revised version of the discussion section that is better organized and more up-to-date.

Reviewers' comments:

Comments to the Author

1. Is the manuscript technically sound, and do the data support the conclusions?

Reviewer #1: Yes

Reviewer #2: Partly

Reviewer #3: Yes

Reviewer #4: Yes

Reviewer #5: Yes

2. Has the statistical analysis been performed appropriately and rigorously? 

Reviewer #1: Yes

Reviewer #2: Yes

Reviewer #3: Yes

Reviewer #4: Yes

Reviewer #5: Yes

3. Have the authors made all data underlying the findings in their manuscript fully available?

Reviewer #1: Yes

Reviewer #2: Yes

Reviewer #3: Yes

Reviewer #4: Yes

Reviewer #5: Yes

4. Is the manuscript presented in an intelligible fashion and written in standard English?

Reviewer #1: Yes

Reviewer #2: Yes

Reviewer #3: Yes

Reviewer #4: Yes

Reviewer #5: Yes

5. Review Comments to the Author

Reviewer #1: A very well done project.

We warmly thank the reviewer.

I would suggest adding information related to other perinatal outcomes that have already been shown to increase in prevalence as a sequela of COVID infection, e.g., preeclamsia (early/tardia), placental abruption.

We thank the reviewer for this suggestion; we have added these data in our manuscript.

Reviewer #2: 1. Was there cross over from those who consented to telemedicine to in-person consult? From the results, it seemed that 100% of the participants had remained on telemedicine follow up. If so, probably can describe in more detail in the discussion why you were able to keep them well managed on telemedicine. How did you pick up the worsening of symptoms. Were you able to remotely measure parameters or examine the patient?

All patients in our study were monitored using telemedicine through the COVIDOM program. We have provided a detailed description of how worsening was detected in the Materials and Methods section.

2. Besides the COVID-19 follow up, was the Ob/Gyn follow up also on telemedicine/hybrid or it was purely in person? 

Ob/Gyn follow-up appointments were conducted in person and were not part of the Covidom process. If patients presented worsening symptoms, they were advised to consult with their own Ob/Gyn.

3. From what i understand, there is a pretty large group 55.7% (pregnant with mild COVID-19) that was on follow up with the health system but did not consent to telemedicine follow up. I thought it would have been useful to compare the 44.3% that had underwent telemedicine guided by the daily self-reporting of symptoms and whether it made a difference compared to those who only came for in-person follow ups when they were unwell, or as a routine.

Certainly, a case-control study would be of great interest. Unfortunately, we do not have data on the cohort that received in-person monitoring to conduct such a study.

4. Specifically for PIH, it did show a significant difference for adverse outcomes as defined by the study. It was not clear COVID-19 worsened the incidence of adverse outcomes or not. From my limited understanding, even without COVID-19 infection, those with PIH could have pre-eclampsia leading to SGA, IUGR, indicated pre-term deliveries already.

Indeed, PIH is an independent risk factor among pregnant women even without COVID-19. Our results indicate that mild COVID-19 does not increase the incidence of adverse events.

Reviewer #3: Thank you for your wonderful efforts and hard work that is clearly reflected in your project. 

I may highlight these points, not to decline your work, but to have a better view of the project outcomes.

I am happy to reconsider your work after making some clarifications on these points:

1- Correlation between the time that the patient had COVID-19 and her gestational age? and was there any difference between different gestational ages and the outcome?

This is a good question. In fact, patients were included at a median of 24 weeks of pregnancy in both groups (those with favorable and unfavorable outcomes).

2- When you describe mild COVID-19, can you please identify all the criteria and clinical symptoms that made you consider it as mild COVID?

We used the term 'mild' because patients were managed in a community setting and did not require hospitalization or oxygen.

3- Duration between having COVID and the outcome. i.e: Can you highlight timelines? if the patient had COVID-19 (mild symptoms) when exactly she either delivered, had a preterm delivery, or had a stillbirth?

We thank the reviewer for its question, we add these data in the result part

4- What do you mean by chest oppression in lines 97 & 138?

We mean the symptom of chest tightness which could be a sign of myocardial infarction or pulmonary embolism. We have modified our manuscript.

5 - Have the patients who had PIH (Pregnancy Induced Hypertension), developed it after COVID-19 or before?

Thank you for this excellent question. We did not collect this particular information, but patients were included in the program when they received a diagnosis of COVID-19, and we collected information on their comorbidities at that time. Therefore, we believe that pregnancy-induced hypertension was present at the time of inclusion and was diagnosed prior to the onset of COVID-19. 

6 - Any of these ladies got vaccinations or not?

Almost none of the patients were vaccinated because the study was conducted during the first three waves of the pandemic, and the vaccine was only available during the third wave.

7- Any other factors that was contributing to the outcome? for example: when you mentioned stillbirth or preterm, do you know if any previous obstetric history of the same issues?

Unfortunately, there was no data on previous obstetric history.

8- Any ultrasound or Doppler studies done for these patients, either before, during, or after having COVID-19, that identified any fetal issues? 

Patients were monitored via COVIDOM and did not have any face-to-face evaluations, so we were not able to perform any ultrasound or Doppler studies.

Reviewer #4: This manuscript seems smart and clear with several meaningful tables. To my disappointed, these data are obtained from self-reported informations, which have many limitations.

For example, in case that pregnant women suffering from COVID-19 infection requires some medications, the kind of drug and the dose may be more influenced to the fetal outcomes.

After reading this manuscript, I have two requests shown as follows;

#1. Although statistic significant difference are shown undoubtedly, the number of fetal death seems much close between hospitalization group (1/102) and non-hospitalization one (5/612). The rate of pregnancy-induced hypertension are not significant differences between these two groups, so I would like you to disclose more information about these six cases such as weeks of delivery and the existence of fetal distress.

We thank the reviewer for this remark. We have added more information about the deaths in our results section. 

#2. In discussion you mentioned the relation between the ICU admission of newborn children and the vaccination of COVID-19. We must desire to know about the relation in this Covidom research. If you have already evaluated the relation, please let us show that.

Unfortunately, nearly none of the patients were vaccinated because we studied patients during the first three surges, and the vaccine was only available during the third surge. 

Reviewer #5: 1. ABSTRACT: The abstract should start with a brief background to the study highlighting what is known or unknown about the proposed study.

We thank the reviewer for giving us the opportunity to improve our manuscript. We have now added a brief background section to the abstract.

Objective: The word “we” should be removed from the objective

We thank the reviewer for their comment and have removed “We” accordingly. 

Methods: The information doesn’t convey succinctly how the study was done.

We thank the reviewer for giving us the opportunity to provide a more comprehensive description of our study in the revised version of the manuscript.

Conclusion: Conclusion should also be made on the role or impact of telemedicine

We have added a paragraph on telemedicine in both the introduction and discussion sections.

2. The introduction is too short and doesn’t state the extent of the problem neither does it contain the justification for the study. A more elaborate introduction should be written.

We thank the reviewer for their comments. We have extended the introduction section to provide more context and justification for our study.

3, The methodology should describe in details what the COVIDOM telemedicine software is all about. How does it work? What version was used for this study? Is it a video conferencing or audio-conferencing app? This information should be clearly spelt out in the methodology

We have more extensively described the monitoring management accordingly. There was no video conferencing involved, but patients who triggered alerts received a phone call and underwent a phone evaluation to determine the best course of management.

4. The methodology should describe in details how the telemedicine was used for the study. How were the questionnaires delivered? Was it delivered to their phones or email? Was there an online/web based face-to-face interview?

We added a more comprehensive description of the tool and the process for managing patients in the COVIDOM cohort.

5 Lines 114,115,116-- Pregnant women with severe COVID-19 had higher risk of preeclampsia, preterm birth, gestational diabetes, and low birthweight than pregnant women with mild disease. This outcome is expected.

We added this remark in the discussion part: “As expected, pregnant women with severe COVID-19 had higher risk of preeclampsia, preterm birth, gestational diabetes, and low birthweight than pregnant women with mild disease”. 

6. Table 2 should be moved to the result section

We moved Table 2 to the Results section.

7. The study involved the use of telemedicine. Was there any advantage regarding the use of telemedicine over other studies that telemedicine was not used? There should be a robust discussion on telemedicine.

A discussion on telemedicine was added to the discussion section.

8. The references were not properly written. Almost all the references still have the DOI numbers. They should be rewritten according to Vancouver guidelines.

We have used the Zotero template for Plos One, which displays the DOI numbers (which is allowed according to the Plos One Submission Guidelines).

---

## [Decision Letter · Decision Letter 1]

6 Jul 2023

Pregnant women with mild COVID-19 followed in community setting by telemedicine, and factors associated with unfavorable outcome

PONE-D-22-18040R1

Dear Dr. Dinh,

We’re pleased to inform you that your manuscript has been judged scientifically suitable for publication and will be formally accepted for publication once it meets all outstanding technical requirements.

Kind regards,

Burak Bayraktar

Academic Editor

PLOS ONE

Reviewers' comments:

Reviewer's Responses to Questions

**Comments to the Author**

1. If the authors have adequately addressed your comments raised in a previous round of review and you feel that this manuscript is now acceptable for publication, you may indicate that here to bypass the “Comments to the Author” section, enter your conflict of interest statement in the “Confidential to Editor” section, and submit your "Accept" recommendation.

Reviewer #2: All comments have been addressed

Reviewer #5: All comments have been addressed

2. Is the manuscript technically sound, and do the data support the conclusions?

Reviewer #2: Yes

Reviewer #5: Yes

3. Has the statistical analysis been performed appropriately and rigorously? 

Reviewer #2: Yes

Reviewer #5: Yes

4. Have the authors made all data underlying the findings in their manuscript fully available?

Reviewer #2: Yes

Reviewer #5: Yes

5. Is the manuscript presented in an intelligible fashion and written in standard English?

Reviewer #2: Yes

Reviewer #5: Yes

6. Review Comments to the Author

Reviewer #2: Thanks to the authors for making the edits. It looks clearer, organised and is much more informative now.

Reviewer #5: All comments have been addressed. However the abstract should start with "Background" and not Objective.

7. PLOS authors have the option to publish the peer review history of their article (what does this mean?). If published, this will include your full peer review and any attached files.

Reviewer #2: No

Reviewer #5: No

---

## [Editor Report · Acceptance letter]

26 Jul 2023

PONE-D-22-18040R1 

Pregnant women with mild COVID-19 followed in community setting by telemedicine, and factors associated with unfavorable outcome 

Dear Dr. Dinh:

I'm pleased to inform you that your manuscript has been deemed suitable for publication in PLOS ONE. Congratulations! Your manuscript is now with our production department. 

Kind regards, 

on behalf of

Dr. Burak Bayraktar 

Academic Editor

PLOS ONE